Delineating the calling pattern of Oecanthus indicus from native and non-native plant species

Sunny Anupam 1
Singh Purnima 2
http://orcid.org/0000-0002-1120-352X Diwakar Swati 2 swati.diwakar@gmail.com
Sharma Gyan Prakash 2 gyanprakashsharma@gmail.com
1 Department of Environmental Studies, Satyawati College, University of Delhi , Delhi , India
2 Department of Environmental Studies, University of Delhi , Delhi , India
Ahmed Nazeer
Electronic publication date: 2023 Oct 18
Publication date: 2023
Volume: 11
Electronic Location ID: e16153
Received 2023 Mar 14; Accepted 2023 Aug 31
Copyright: © 2023 Sunny et al.
Copyright year: 2023
Copyright holder: Sunny et al.
License: This is an open access article distributed under the terms of the Creative Commons Attribution License, which permits unrestricted use, distribution, reproduction and adaptation in any medium and for any purpose provided that it is properly attributed. For attribution, the original author(s), title, publication source (PeerJ) and either DOI or URL of the article must be cited.
License URL: https://creativecommons.org/licenses/by/4.0/

Keywords: Tree crickets, Oecanthus, Lantana, Adhatoda, Hyptis, Host shift, India

Funding: University Grant Commission, India for Junior Research Fellowship Science and Engineering Research Board, Department of Science and Technology, India Purnima Singh received a University Grant Commission, India for Junior Research Fellowship. Swati Diwakar and Gyan Prakash Sharma received financial support from the Science and Engineering Research Board, Department of Science and Technology, India. The funders had no role in study design, data collection and analysis, decision to publish, or preparation of the manuscript.

==============================
The study attempted to understand the effect of the host plants on the call parameters of native tree cricket, Oecanthus indicus (Order: Orthoptera, Sub-order: Ensifera, Family: Gryllidae) while calling from native (Justicia adhatoda) and non-native host plant species (Lantana camara and Hyptis suaveolens). The study was conducted at four locations across India. Calls of O. indicus were recorded on these host plants in the field and spectral and temporal parameters of calls were analysed. The results suggested that the peak frequency varied among the two non-native plant species while the difference in temporal pattern between the native and non-native host plants was observed only in the syllable period. The study also quantified the choice of calling positions of insects from the three-host species. The native O. indicus chose non-native H. suaveolens leaves extensively as a preferable site to baffle (37%). Differences in the call parameters and choice of the host plant by insects may ultimately affect the preference and performance of insects on invasive plants. The study would aid in exploring the underlying evolutionary and ecological processes of adaptive success of insects on non-native plants.

Introduction

The introduction of the non-native plant species in a novel environment is often accompanied by significant impacts on the native flora and fauna. The invasive species tend to outcompete the native vegetation and cause alteration in the existing plant-insect interactions thus affecting their diversity, abundance, biomass, richness, and species composition (Castells et al., 2014). The spatial dominance of a plant species depends on the temporal occurrence of the non-native and native plant species in an area. The new plant-insect associations that are established under natural conditions are constrained by the spatial and temporal distributions of the insects and plants. Such interactions are subjected to varied ecological and evolutionary phenomena such as ecological fitting and evolutionary traps (Janzen, 1985; Schlaepfer, Runge & Sherman, 2002). When insects encounter non-native plants, they may be attracted to them initially due to several traits that resemble their preferred native host plants. However, these non-native plants may have detrimental effects on the insects’ survival and reproduction due to a lack of essential resources or the presence of toxic compounds. In such cases, the insects’ innate preferences or behaviors can become “trapped” or mismatched with the novel environment, leading to reduced fitness over time, referred to as evolutionary traps (Schlaepfer, Runge & Sherman, 2002). Moreover, the ecological interactions between organisms could arise due to their matching ecological traits and environmental factors at play. Agosta & Klemens (2008) proposed that “phenotypic plasticity, correlated trait evolution and phylogenetic conservatism” act as means of ecological fitting. Non-native plants may initially render themselves as a suitable habitat due to their resemblance to the habitats of insects based on various physical properties, chemical or visual cues, nutritional resources, phenological synchronizations, etc. The non-native plants often provide alternative food resources to the local herbivores. Host switching may be advantageous as it serves as a complementary food source (Agosta, 2006). For example, several species of butterflies have been reported to oviposit or feed on exotic plants that ultimately help them in the expansion of their geographical range and breeding seasons (Graves & Shapiro, 2003). Host preference and eventually the performance of insects on abundant non-native plants are greatly governed by the above-mentioned processes. Host shifts are dependent on a species’ host preference functions when native and non-native hosts co-occur in the same habitat (Castells et al., 2014).

The ecological role of insects in terms of establishment, colonization, and naturalization of invasive plants through interactions such as herbivory, seed dispersal and pollination has been extensively studied (Pearse & Altermatt, 2013; Bezemer, Harvey & Cronin, 2014; Sunny, Diwakar & Sharma, 2015). However, there remains a dearth of studies focusing specifically on acoustically active insects and non-native plants as their hosts.

Among the soniferous insects, crickets (Order: Orthoptera) are widely spread across the tropics and have reasonably high abundance and distribution. Orthopteran insects produce diverse and species-specific sounds. They use acoustic communication for interspecific interaction, intrasexual competition, and territorial defense (Hall & Robinson, 2021). The tree cricket genus Oecanthus produces song through stridulation, by rubbing the specialized tegmina against each other. The sound is produced during the closing stroke of the wing motion (Walker, 1962). Oecanthus uses acoustic communication for species recognition and mate attraction (Metrani & Balakrishnan, 2005). Insects like O. indicus may prefer native or non-native plants as a site for feeding, mate attraction using acoustic signals, and mating. “Preference” is the ability of an insect to choose a host plant for food or oviposition. The insect may tend to choose similar habitats despite experiencing reduced fitness or performance (Schlaepfer, Runge & Sherman, 2002). ‘‘Performance’’ is quantified based on the total number of eggs laid and their larval development (Sunny, Diwakar & Sharma, 2015). Thus, host preference could be an important attribute that affects the reproductive fitness of an acoustically communicating insect. Studies have reported the association of Oecanthus species with a non-native plant, H. suaveolens (Bhattacharya, Isvaran & Balakrishnan, 2017). The association of Oecanthus species with non-native plants can provide an excellent system to investigate the effect of preference based on various call characteristics.

The suitability of the habitat is an important parameter that can affect reproductive success in terms of male fitness and call quality. In terms of acoustically communicating crickets, spectral and temporal features are used by conspecific females to assess male call quality (Greenfield, 1997; Symes, 2018). It has been observed in field crickets, that females prefer a male calling song with a higher call rate and longer call duration (Wagner, 1996). Call rates are known to be affected by diet and male nutrition (Wagner & Hoback, 1999). Andrade & Mason (2000) found that male cricket Ornebius aperta that fed on a high-nutrient diet were healthier and transferred more spermatophores on average than low-diet males.

Reproductive success and survival in crickets are also dependent on their calling sites. These sites help them in predator avoidance and influence their probability of securing mates. They also manipulate the plant leaves by making holes in the leaf and calling from them for sound amplification (Deb, Modak & Balakrishnan, 2020).

Studies have investigated the effects of host plants on insect performance in terms of oviposition success, larval development & performance, adult body mass, and fecundity (Keeler & Chew, 2008; Fortuna et al., 2013) while nothing is known about the effect of host plant and calling positions on the calling parameters of acoustically active species. There are no bioacoustics studies where the comparative effect of native and non-native host plants is studied. We studied the acoustic parameters of tree cricket O. indicus across native and non-native host plant species.

The objectives of the study were to investigate (i) variation in call parameters of O. indicus calling from native and non-native plants and (ii) the choice of calling position of O. indicus from native and non-native plant species. We tested the hypothesis that there is no difference in the call parameters between O. indicus calling from native and non-native plants.

Materials and Methods

Study site and period

The sampling was carried out from 2013 to 2015 across four different locations in India viz. Dehradun (30.2333°N, 78.1667°E); Delhi (28.6°N, 77.18333°E); Dhanbad (23.7957°N, 86.4304°E) and Muzaffarpur (26.1209°N, 85.3647°E) (Fig. 1). Sampling locations were selected based on the presence of O. indicus in dominant stands of native plant, J. adhatoda and non-native, H. suaveolens and L. camara. The temperatures ranged from 17.5 °C to 33.5 °C while the relative humidity ranged from 55% to 98% across the sampling locations.

Figure 1 Map showing the study locations in India.

Study system

The tree cricket genus Oecanthus belongs to the sub-order Ensifera and family Gryllidae of the insect order Orthoptera. These crickets are small-sized, nocturnal, and semi-arboreal, widely distributed across the ecoregions of the world except the poles. The genus Oecanthus has four described species (O. rufescens Serville, O. henryi Chopard, O. bilineatus Chopard and O. indicus Saussure) on the Indian subcontinent (Metrani & Balakrishnan, 2005). O. indicus is a generalist, widely distributed and observed to be calling from both native and non-native plants. The male body size of O. indicus generally ranges between 15.32 ± 1.03 mm (Metrani & Balakrishnan, 2005).

H. suaveolens (Lamiaceae) is native to Tropical America but is now found across the globe and emerged as a pantropic weed (Sharma, Raizada & Raghubanshi, 2009). It infested the natural areas of the Vindhyan highlands in the latter half of the 20th century (Agrawal, 2002). It is one of the abundantly spread species of the Vindhyan highlands forests of India after Lantana camara (Sharma, Raizada & Raghubanshi, 2009). A significantly large portion of the total geographic area of India i.e., approximately 40.20% (1,320,119 km2) is predicted to be suitable for Hyptis (Padalia, Srivastava & Kushwaha, 2014). So, the species has a likelihood of rapid spread and subsequently its interacting insect species.

Lantana is a member of the family Verbenaceae, it is native to Tropical America. Lantana was first introduced to India from 1807 onwards. It perpetuated throughout the country during the colonial rule. Lantana camara has its first archival record from 1891 in India (Kannan, Shackleton & Uma Shaanker, 2013). It is naturalized in India and occurs as dense monospecific thickets. Owing to its rapid naturalization and invasion outside its native range, it is considered a weed of international significance (Sharma, Raghubanshi & Singh, 2005). Using extensive field sampling data and modeling, Mungi, Qureshi & Jhala (2020) have reported that 44% of the Indian forests are invaded by L. camara.

J. adhatoda is a native to India belonging to the family Acanthaceae. It is a perennial, evergreen, and highly branched shrub widespread throughout the tropical regions of Southeast Asia (Dhankar et al., 2011). J. adhatoda has been found to co-occur with the other two non-native plant species.

Acoustic sampling

O. indicus calls were recorded in the evening between 1,900 and 2,100 h at the study locations. Insects were psychoacoustically located and sound recordings of 30 s were taken using a digital recorder (TASCAM DR-08, TEAC; America Inc., Newtown Square, PA, USA, 44.1 kHz, 16 bits, .wav format) from a distance of 25 to 50 cm. The distance was maintained in the range of 25 to 50 cm from the calling insect to maintain a balance between avoiding near-field effects and to achieve a better Signal to Noise ratio (SNR) for signal analysis.

Ambient temperature and relative humidity were recorded using a pocket weather meter (Kestrel 4500). We recorded 16 individuals from Dehradun, three from Dhanbad, 29 from Delhi and two from Muzaffarpur. Representative calls from different host plants are provided (see Supplementary Material). Although O. indicus has been reported to be distributed from the entire country, most of the call recordings are from Dehradun and Delhi owing to the constraints of carrying out field work at night due to various factors such as permits, safety concerns, suitable sites away from anthropogenic disturbances, availability of invasive plant species etc. We believe that location bias in the call recordings from the two locations would be minimal (Walker, 1962) as both lie in the Northern plains of the country and have roughly similar climatic conditions. Calling positions refers to the part of the plant from where it called (i.e., top of the leaf, between two leaves, baffles, and leaf margin).

Acoustic analyses

Spectral and temporal patterns of the sound recordings of O. indicus, such as peak frequency, echeme duration, echeme period, syllable duration, and syllable period were analysed. The reported call parameters were regressed to 22 °C (the mode at which most calls were recorded) to avoid any confounding effects that temperature might have on call parameters (Metrani & Balakrishnan, 2005) (Table S1). In a study on the stridulatory rates in bush crickets, Walker (1975) revealed that the effects of humidity are of little significance under field conditions hence humidity was not considered in the analysis. The call parameters of individuals calling from different plant species were compared only with the regressed values of the call parameters. Peak frequency (PF) is the frequency with the highest amplitude. Syllable represents the sound produced by a complete stridulatory movement (during the closing stroke of the wings), syllable duration (SD) being the time period from the beginning to the end of a syllable. Syllable period (SP) is the time period from the beginning of a syllable to the beginning of the next. Echeme represents the sound produced by the multiple, subsequent opening-closing movements of the wing and is the first-order assemblage of syllables. Echeme duration (ED) is the time period from the beginning to the end of an echeme. Echeme period (EP) is the time period from the beginning of an echeme to the beginning of the subsequent one (Baker & Chesmore, 2020) (Fig. 2). Individual recordings with a sampling rate of 44.1 kHz were analysed at FFT size 1,024 in the Hanning window. We analysed 25 echemes per recording.

Figure 2 Schematic representation of the call parameter terminologies following Baker & Chesmore (2020).

RAVEN Pro 1.4 (Cornell Lab of Ornithology, Ithaca, NY, USA) and Spectra Plus 5 (Pioneer Hill Software, Poulsbo, WA, USA) were used for temporal and spectral analysis.

Statistical analyses

Non-parametric statistics were performed as the data was not normally distributed. Kruskal-Wallis ANOVA was used to determine differences between the call parameters. The coefficient of variation (CV) for call parameters on each host plant was calculated to compare the degree of variation between datasets. Post-hoc test was carried out to examine the differences in plant species. All the statistical tests and analyses were conducted using StatSoft Inc (2013) and Systat (2021).

Results

Effect of host plant on acoustic parameters

We analysed a total of 50 recordings of O. indicus (one recording per individual). A total of 14 individuals calling from native J. adhatoda, 19 from non-native L. camara and 17 from H. suaveolens plant species were recorded. The peak frequency (kHz) of O. indicus calling on J. adhatoda (2.34 ± 0.17 kHz, CV = 7.42%) was similar to O. indicus calling from L. camara (2.43 ± 0.17 kHz, CV = 7.35%). The peak frequency of O. indicus calling from H. suaveolens was lower compared to the other two host plants (2.26 ± 0.19 kHz, CV = 8.19%). Despite having low CV values, peak frequency of O. indicus individuals calling from L. camara was significantly different from individuals calling from H. suaveolens (Kruskal-Wallis, H (2, N = 50) = 7.24, p = 0.027) (Fig. 3A) (Table 1).

Figure 3 Box-whisker plots showing the temporal and spectral parameters i.e., (A) peak frequency (kHz), (B) syllable period (ms), (C) syllable duration (ms), (D) echeme period(s), and (E) echeme duration(s) of the three host plant species.

The bar denotes median, box shows the quartile range (25–75%), and whisker denotes the non-outlier range. The asterisk indicates a significant difference.

Table 1 Results of Kruskal-Wallis ANOVA test of call parameters (dependent variables) between host plant species (grouping variable) and post-hoc comparisons (z′ values are quoted).

Call parameters	Host sp.	L. camara	H. suaveolens	
PF (kHz)	L. camara	-	–	
(H = 7.24,	H. suaveolens	2.69	–	
p = 0.027)	J. adhatoda	1.20	1.31	
SP (ms)	L. camara	-	-	
(H = 6.36,	H. suaveolens	1.37	–	
p = 0.041)	J. adhatoda	2.51	1.18	
SD (ms)	L. camara	–	–	
(H = 0.54,	H. suaveolens	0.01	–	
p = 0.76)	J. adhatoda	0.65	0.65	
ED (s)	L.camara	–	–	
(H = 2.07,	H. suaveolens	0.62	–	
p = 0.36)	J. adhatoda	0.88	1.44	
EP(s)	L. camara	–	–	
(H = 5.79,	H. suaveolens	1.85	–	
p = 0.06)	J. adhatoda	0.54	2.24	
Note:

Peak frequency (PF), syllable period (SP), syllable duration (SD), echeme duration (ED) and echeme period (EP) (n = 50). Bold letters denote statistical significance at p < 0.05.

The syllable period ranged from 29.62–42.91 ms between the three host plant species. However, the syllable period of O. indicus calling from native J. adhatoda (42.91 ± 23.31 ms) was significantly different from non-native L. camara (29.62 ± 9.17 ms) (Kruskal-Wallis, H (2, N = 50) = 6.36, p = 0.041) (Fig. 3B) (Table 1). Syllable duration was found to be between 17.25–17.95 ms on all plant species. There was no significant difference in the syllable duration between individuals calling from L. camara (17.25 ± 3.54 ms), H. suaveolens (17.45 ± 1.46 ms), and J. adhatoda (17.95 ± 2.29 ms) (Kruskal-Wallis, H (2, N = 50) = 0.54, p = 0.76) (Fig. 3C) (Table 1).

The echeme period of O. indicus individuals ranged from 0.95 ± 0.33 s on J. adhatoda to 1.23 ± 0.37 s on H. suaveolens respectively. There was no significant difference in the echeme period of O. indicus calling from the three plant species (Kruskal-Wallis, H (2, N = 50) = 5.79, p = 0.06) (Table 1, Fig. 3D).

The echeme duration of O. indicus individuals calling from L. camara (0.77 ± 0.42 s), H. suaveolens (0.75 ± 0.26 s), and J. adhatoda (0.66 ± 0.35 s) were not significantly different (Kruskal-Wallis, H (2, N = 50) = 2.07, p = 0.36) (Table 1, Fig. 3E).

The use of calling position from native host plant vs non-native host plants

O. indicus was found to be calling from four positions on native and non-native species viz. top of the leaf, leaf margin, using the leaf as a baffle, and between two leaves (Fig. 4). Leaf margin was found to be the most frequently used position by O. indicus on L. camara (60%) and J. adhatoda (60%) (Fig. 4). Using leaf as baffle was predominantly seen on H. suaveolens (36.84%) and then on J. adhatoda (20%). O. indicus also called from the top of the leaf on H. suaveolens (42.10%) (Fig. 4).

Figure 4 Calling positions (%) of O. indicus on native and non-native plants.

L. camara (n = 20), H. suaveolens (n = 19) and J. adhatoda (n = 15).

Discussion

The study attempted to understand the interactions between insects and non-native plants through changes in the call parameters of an acoustically communicating insect species. The results of the study showed that the peak frequency of O. indicus calling from L. camara was significantly different from that of non-native H. suaveolens and the syllable period of O. indicus calling from L. camara was significantly different from that of native J. adhatoda.

Temporal patterns such as echeme duration, echeme period, and syllable duration did not show any difference between individuals calling from native and non-native plants. Studies have shown that frequency component of songs are directly controlled by emitting structures whereas temporal structures are more variable (Alexander, 1962; Walker, 1962). Variable temporal structures are more likely shaped by environmental and genetic differences between individuals. However, such variations are considered as minor (Walker, 1962). Studies have investigated the effect of diet on the reproductive fitness and growth performance of orthopterans (Magara et al., 2019). Several studies have highlighted the effect of diet on male signal structure. As in the case of Gryllus lineaticeps, males produced more attractive signals when they were fed with higher-quality diets. Males on high-nutrition diets called significantly at higher rates than those fed on low- nutrition diets (Visanuvimol & Bertram, 2010).

Although plants may not directly change characteristics of the calls produced by crickets as the call parameters are primarily determined by the physical properties of stridulatory organs involved in sound production. Not much work has been carried out to study the effect of host plants on spectral and temporal parameters of cricket or bush crickets song. However, it has been shown that the host plant quality affects the performance and reproductive strategies of insects and their interactions (Awmack & Leather, 2002). Moreover, various nutrients available in plants through diet can affect the cricket’s physical condition and overall health, which, in turn, can influence its ability to produce sound (Visanuvimol & Bertram, 2010). The diet of the crickets affects their condition as well as important fitness traits such as body size and lifetime reproductive success. The acoustic mate attraction signals are fairly dependent on cricket body size and condition (Visanuvimol & Bertram, 2010). It can be speculated that different plants can provide differential dietary nutrients for the development of the tree cricket and subsequently, the peak frequency of its sound. The metanotal gland feeding, a nuptial gift given by singing males to females, is largely dependent on the male diet (Smith et al., 2017). Comparative quantification of nuptial gifts by O. indicus feeding on native and non-native plants could further provide insights into the role of non-natives as evolutionary traps.

The change in call parameters exhibited by tree crickets associated with these plant species could be an indicator of performance differences between invasive and native plants. Ongoing preference-performance experiments in the laboratory involve measuring and weighing lab-reared individuals. Additionally, controlled experiments are being conducted, where tree cricket specimens would be grown on different plant species, providing insights to the current study. Such controlled experiments would evaluate the interactions between tree crickets and specific plant species by controlling many confounding factors such as feeding preferences, nutrient availability, and the size of the animal to infer the performance and/or fitness levels of these insects in relation to invasive vs native plants.

The calling position of O. indicus varied as it chose non-native H. suaveolens leaves extensively as a site to baffle (36.84%) as compared to other plant species (Fig. 4). The study showed that the cricket species exhibited more baffling behaviour on leaves of H. suaveolens. Oecanthus use baffles as a strategy to amplify signals for effective long-distance communication. It is a reproductive strategy where the males, often the smaller and low-amplitude callers have resorted to this strategy to acquire mates (Deb, Modak & Balakrishnan, 2020). Building a baffle would have clear benefits for the individual singer in terms of mate attraction, so a lack of observed baffle building behaviour on one non-native host plant and an increase in the same behaviour in another non-native host plant could have very interesting consequences for both host plants and tree crickets. This is suggestive of a preferential host shift for reproductive success. Shift to novel hosts can have implications on various traits of the insects. Cocroft (2007) opined that the host plant environment as a means of production, transmission, and propagation of signals is very crucial for sexual communication, mate recognition, and attraction. Host shifts may drive natural selection through divergence and signal evolution of acoustically communicating insects and are often associated with ecological speciation in plant-feeding insects (McNett & Cocroft, 2007).

O. indicus is predominantly found in shrub lands/wastelands and distributed throughout India with no/limited geographical isolations between them. Population differences in separate geographical locations are a long-term and cost-intensive study. We plan to investigate the geographical or population-based differences in future.

Conclusions

The study revealed significant differences in the peak frequency of O. indicus calling from the two non-native plant species-L. camara and H. suaveolens while the syllable period varied significantly between the calls from the native plant species J. adhatoda and non-native plant species L. camara. In terms of the calling positions, calling through the holes in the plant leaves i.e., baffling was extensively observed in H. suaveolens. The top of the leaf was the second preferred calling position by O. indicus. Empirical studies investigating the calling parameters of Oecanthus from various calling positions on native and non-native plants are needed to further shed light on the possibility of host shift of Oecanthus on non-native plants. Future work evaluating the diet preference, mating success, and reproductive output of Oecanthus species on native and non-native plants will help provide insights into the preference and performance of insects found closely associated with non-native plant species.

Supplemental Information

Supplemental Information 1 Justicia adhatoda.

Click here for additional data file.

Supplemental Information 2 Lantana camara.

Click here for additional data file.

Supplemental Information 3 Hyptis suaveolens.

Click here for additional data file.

Supplemental Information 4 Supplementary table.

Click here for additional data file.

We thank Dr. Chandranshu Tiwari and Karuna Gupta for their help in the call analysis. We are thankful to the reviewers for their insightful and valuable comments on the manuscript.

Additional Information and Declarations

Competing Interests

Author Contributions

Data Availability

The authors declare that they have no competing interests.

Anupam Sunny conceived and designed the experiments, performed the experiments, analyzed the data, prepared figures and/or tables, authored or reviewed drafts of the article, and approved the final draft.

Purnima Singh analyzed the data, prepared figures and/or tables, authored or reviewed drafts of the article, and approved the final draft.

Swati Diwakar conceived and designed the experiments, analyzed the data, prepared figures and/or tables, authored or reviewed drafts of the article, and approved the final draft.

Gyan Prakash Sharma conceived and designed the experiments, analyzed the data, prepared figures and/or tables, authored or reviewed drafts of the article, and approved the final draft.

The following information was supplied regarding data availability:

The raw data is available in the Supplemental File.

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
