# Peer review of "Delineating the calling pattern of Oecanthus indicus from native and non-native plant species"

_PeerJ, doi:10.7717/peerj.16153_

## Round 0.1 · original submission · Major Revisions

The manuscript entitle" elineating the calling pattern of Oecanthus indicus from native and non-native plant species" written by Sunny et al is well organised as well as high quality figures and added relevant references. But still its need improvement in several parts mentioned by reviewers. I am accepting this manuscript after major revision modification according to reviewers comments

·

Basic reporting

The manuscript „ Delineating the calling pattern of Oecanthus indicus from native and non-native plant species” by Sunny et al. deals with bioacoustics of insects in relation to native and non-native plants. The authors analysed calling characters of a species of tree crickets and indicate that some calling parameters might change when the species uses non-native plants as calling site.
The idea of studying bioacoustics in relation to climate change and invasive plants is certainly worth the be considered and studied. The authors report that peak frequencies (PF) of the calls and temporal pattern (syllable period, SP) of the species depend on the plant species. However, the data basis is relatively small (n= 15-20) and, more importantly, additional parameters have not been included in the analysis. Temperature, humidity, size of the animal, singing site and location might have a large impact on the call parameters. The manuscript lacks any explanation, why the call parameters have changed.

The manuscript is well organised. It includes the relevant background. Literature is well referenced and relevant. The figures are of high quality and illustrate the findings (although the data of figure 2 are essentially the same as in table 1 and in the text). The raw data are presented as excel file, whereby only sound characters are given. No recorded sound tracts are made available, nor do the sheets include other monitored data, like temperature or singing position.

Experimental design

The experimental design is adequate, but circumstances need to be reported carefully. The research question is new and interesting. The methods are described in sufficient detail, but, as mentioned above, not included in the raw data. Data on the structure of habitats are missing, e.g. data on the plant structure nearby the recording. Such data might be important for analysis of sounds in respect to frequency filtering in the habitat.

Validity of the findings

The findings are novel and supported by the statistics. However, there are some problems with the results.
The change in peak frequency would imply that the plant has influence on the peak frequency.The spectral content is generated at the wings of the cricket. The plant only assists radiation of the sound wave. How could the plant change peak frequency?
The habitat can act as a high-pass frequency filter, but data to the habitat structure are not given. It is only reported in MM that the distance was at least 25cm. What was the maximum distance?
Data on the size of the animals are missing. Size might strongly influence behaviour and calling behaviour. Additionally, it would be interesting to know whether the animals have also fed during the nymphal stages on the plant, where they were singing.
The temporal data (syllable period, chirp duration etc) strongly depend on the temperature. The authors mention that temperature has been recorded, but do not provide these data or include them in the statistical analysis.
How could a plant influence one, and only one, temporal parameter of the song?
A principle components analysis of all data (sound parameters, temperature, calling site, location, body size, wing size etc) is needed to validate the findings.

Additional comments

Comments to the manuscript
Introduction: it would be nice to get some information about the host plants. E.g. is it known how long the non-native plants are in India, respective the recording locations?
L 66-71: generally, the statements are correct, however, they are based on different species. Is it known, whether Oecanthus females uses call parameters to assess sperm quality (see in this context Deb et al. 2012)?
L 75 sound is not amplified, but energy loss is reduced.
L 76 ff Probably no bioacoustic studies comparing native and non-native plants are published. However, for this comparison it might be helpful to include studies in which insects use different (native) host plant species or other different calling conditions – I am not aware of a study, that reports different PF or SP in correlation to such conditions.
Methods: Temperature ranged from 17.5° to 33.5°C, e.g. by a factor 2x! This factor could account for the differences in SP. Please include these data to each single recording and in the analysis. The same might be true for the relative humidity, which can influence frequency dependent attenuation.
L 96ff: the information about other species is not needed.
L 123: it is of importance to report not only the minimum recording distance, but the actual recording distance.
Results: The figures of PFs are given in the text, in a table and in a figure. This can be reduced. The same holds for the SP.
L 172 ff: it is important to report and correlate the singing sites to the call data.
Discussion: A sequence of correct, but somewhat isolated statements. Possible reasons of the findings of variation in PF and SP of the calls are not discussed.
Figure 1: it is difficult to identify the study sites on the map. Why have the states of India different shadings?
Figure 2: Include line to indicate the significance pairs.
Figure 3: include N in the legend. The Y-axes should be 100%
Table 1: might not be necessary, as the data are reported in the text and figure 2.
Table 2: Why is the column J. adhatoda empty?

Reviewer 2 ·

Basic reporting

English
I found several words or lines that are difficult to apprehend, because the English language is not very clear or understandable. I marked those in the pdf. Please consult a fluent speaker or English tutor. Below some specific improvements, as examples.
In line 28, you describe a “dynamic transition of native plant and non-native plant abundance is taking place” and variants of such wording in line 35. I suppose you are trying to describe the process of infestation with non-native plants. Please use a more clear description of this process.
The plural of phenomenon is phenomena. Line 36 and 44
Line 154: please use “was lower compared to…” instead of “was lesser as compared to…”

I think literature references are fine, but focussed on the text, that really needs improvement.

Article structure is fine, including data

Results are relevant in the light of the scope of the research. However, part of the supposedly collected results on temperature conditions have not been included in the analysis.

Experimental design

Original research, however not formulated in full clarity. Might have filled an identified knowledge gap, but discussion and conclusion fail to give a full overview of factors that may explain the results. When taken into account, other factors may explain the results better than concluded in the article. Several comments supporting this opinion are given under 3 and 4.

Validity of the findings

There seems not to have been room for alternative explanations for the results, which have only be presented in the light of the hypothesis. Although the hypothesis is a interesting en potential relevant one, the findings of this research are potentially not useful. Supposedly another experimental setup with e.g. a standardisation of temperatures and transplantation of specimens from one plant species to another plant species would be possibilities to improve de outcomes with respect to the hypothesis.

Additional comments

Although you mention recording temperature (and humidity) in Materials & Methods, there is no mentioning of temperature in the results, discussion, or conclusions. Also, in the data sheet temperature data are missing. As you may well be aware, Syllable duration and syllable period are dependent of the ambient temperature in crickets. Even peak frequency may vary with temperatures. You mention a range of 17.5-33.5℃, which is considerable. I would not be surprised if the variation you found in the measured parameters is mainly caused by differences in temperature. At least I suppose it will blur your results considerably. Before publication, I would suggest performing a proper analysis of the effects of differences in temperature.

Based upon the morphology of the wings, I expect the species can fly. It would be able to fly to other plant species. If your results proof right, it may sing differently on a different plant species. In my opinion, the ecological and evolutionary impact would be much smaller than you are suggesting in the article if just flying to another plant species would yield some variation in the song characters.

In several instances, you use the term host (host plant, host species, host preference) for the plant species you investigated. However, Oecanthus species usually use (many) different plant species as food and as singing post. They are usually not very dependent of one plant species for completing their life cycle as is the case with typical monophagous insects. Is O. indicus much different in this respect and more or less confined to just these three plant species?

You are using the term calling/singing site (or microhabitat site), but based on the context, I think you mean calling post or exact calling post or exact calling position. These latter terms would be the terms that are also used for birds. Site more often describes a potentially wider area, like a vegetation, parcel, or area.

In line 65 you refer to “females refer a male calling song…”. This looks like a general statement. My comment here is about specificity. The investigation by Wagner (1996) specifically refers to one species of field cricket (Gryllinae), not to Oecanthinae. I would suggest to either be more specific, stating this has been investigated in a field cricket, or add more references to make this a more general statement.

A minor point: the term chirp is not very often used for a first-order assemblage of syllables. More usual (and advised) is the term echeme. See: Baker E, Chesmore D (2020) Standardisation of bioacoustic terminology for insects. Biodiversity Data Journal 8: e54222. https://doi.org/10.3897/BDJ.8.e54222

Annotated reviews are not available for download in order to protect the identity of reviewers who chose to remain anonymous.

·

Basic reporting

The introduction of the manuscript, whilst written in good professional English, suffers from a general lack of focus and structure. In essence, a clearer structure within the introduction's "narrative" and well delineated aims would greatly enhance the manuscript. Specifically, I would encourage the authors to start with a broader introduction to invasive plant/native insect interactions and their potential implications for ecology and evolution, including some examples from the literature. this could be followed by the general observation that the effects of invasive species as potential host plants for acoustically active insects are poorly known, before explaining the specific system of acoustic signalling in crickets/bush crickets and especially in the tree crickets in more detail (and with more references to the current literature). Knowing about the communication system of the crickets, one can then proceed to form a rationale as to why the choice of host plant can have positive or negative fitness effects on both individual and the population/species as a whole, leading to the formulation of clearly delineated aims and/or hypotheses.
Please see also my comments directly in the manuscript PDF.

Experimental design

I find the general question of the manuscript, namely if an insect's choice between native or non-native host plants has an effect on key acoustic parameters of its communication signals, highly interesting. However, the authors have not designed their experiments and statistical analyses in a way that allows for finding an answer to this particular question.
Song parameters in Oecanthines are strongly dependent on temperatures (Walker, 1962; Metrani et al., 2005; Mhatre et al., 2011 & 2012). This includes peak song frequency and syllable durations. As you do not report recording temperatures for each recorded call and only state a general temperature range spanning more than 15 °C, your statistical analyses and the interpretation of those results are meaningless. All variation seen within the data could easily be explained by temperature differences between the recording sites or even between the recorded individuals. Additionally, recording sites are not factored into your statistical analysis. Since recording sites are located many hundreds of kilometres apart from each other, some of the variations seen in song parameters could also be attributed to population differences, especially when local populations have been geographically isolated for some time.
Furthermore, it is unclear if you recorded 50 individuals and analysed many calls per individual, if you just present data of 50 calls from any number of individuals or if you analysed one call per individual for 50 animals. For a meaningful data analysis, an equal number of calls per individual should have been analysed, data for each parameter averaged per individual and then data for all individuals and host plants can be compared. However, this is again only true for songs recorded at the same temperatures and preferably at the same sampling site.
In essence, you neglected to include two major confounding factors in the statistical analysis, temperature and sampling site, leading to results that cannot be interpreted meaningfully.
In general, I would suggest to start studying the effects of host plants on call parameters in a more controlled lab-environment, with stable temperatures and a well-characterised population of study animals.

Validity of the findings

See my comments above and in the PDF.

In addition, I would like to mention here that the findings concerning the apparent differences of singing location and of baffle building between host plants (however, no Ns reported here) is intriguing. Building a baffle should have clear benefits for the individual singer in terms of mate attraction, so a lack of observed baffle building behaviour on one non-native host plant and an increase in the same behaviour in another non-native host plant could have very interesting consequences for both host plants and crickets. This could have been also expanded on and discussed more broadly in the discussion section.

Additional comments

See my comments above and in the PDF

---

## Round 0.2 · Minor Revisions

The manuscript entitle"Delineating the calling pattern of Oecanthus indicus from native and non-native plant species" written by Sunny et al is well organised as well as high quality figures and added relevant references. Authors did well and modified the manuscript as per reviewers' comments.

Please address the remaining minor revisions.

·

Basic reporting

The manuscript „ Delineating the calling pattern of Oecanthus indicus from native and non-native plant species” by Sunny et al. deals with bioacoustics of insects in relation to native and non-native plants. The authors analysed calling characters of a species of tree crickets and indicate that some calling parameters might change when the species uses non-native plants as calling site.The idea of studying bioacoustics in relation to climate change and invasive plants is certainly worth the be considered and studied.
The authors carefully replied to my (and other) comments of the first version of the manuscript (and I am sorry, for my error, describing habitats as high-pass filter). The text is much improved and the results are discussed adequately.

Basic reporting
The revised manuscript is improved and well organised. It includes the relevant background even better. The figures are improved and illustrate the findings. The raw data are presented as excel file, allowing secondary analysis – see Validity of findings.

Experimental design

Experimental design
The experimental design is adequate, and circumstances are now reported. The research question is new and interesting. The methods are described in sufficient detail. However, the analysis should include more details (see Validity of findings). It would be nice to a larger sample with more homogenous data distribution. For example, the data might have a location bias, as recordings have mainly be done in two locations, with different plant species.

Validity of the findings

The findings are novel.
From the extended data sheet, it becomes obvious that the authors used a regression to calculate the peak frequency for 22 degrees temperature. What is the basis of this regression? Please, give a reference on how peak frequency is changing with temperature. To my opinion, temperature has in this range only a minor effect on peak frequency and the uncorrected data should be used.
The authors report an n=19 for recording at L. Camara. However, it seems that they have excluded two recordings with untypically high peak frequency. Thus, correct the n to 17 and please report, what might be the reasons for the high peak frequencies and why they have been excluded.
Furthermore, as is now obvious, most recordings are from just two locations, DDN and DLI. Comparing the peak frequencies from these two locations result in a P value of 0.0074 (T.Test, treated as two independent samples). Together with the non-homogenous distribution of host plants, this could indicate that the differences in peak frequency is due to local populations, rather than to plant species. I would suggest more statistical data analysis (this is why small sample size might be problem; more variables than the plant species can influence the data).
Similarly, the basis of the regression used for the temporal parameters should be stated to detail in MM.

·

Basic reporting

The basic reporting has been substantially improved from the first manuscript submission, including a restructuring of the introduction, the figures and the discussion. Issues raised in the first review round have been addressed adequately.

Experimental design

The Materials & Methods section has been expanded with some welcome details and the addition of the missing temperature regression has additionally improved this section and the following results section majorly.

Validity of the findings

The reviewers main concern about the results, namely the influence of different recording temperatures on calling song spectral and temporal structures, where addressed in the data analysis and additional data showing regression of the parameters to a standard temperature of 22 °C was submitted. The data and statistical analysis is thus now more robust than before and results can be interpreted accordingly.
The results now do indeed point towards a difference in song peak frequency for animals singing on H. suaveolens plants and potential reasons for the differences are discussed.

Additional comments

The authors have performed major revisions on the manuscript and improved it substantially, leading to the presentation of intriguing results on potential changes in the song parameters and singing behaviour of tree crickets in relation to their choice of native or non-native host plants. The authors have hinted in the discussion that more controlled laboratory experiments are being conducted on the influence of host plants on the acoustic behaviour and I'm looking forward to reading the results of those in the future as these should shed some light on potential mechanisms responsible for the observed changed in song frequencies.

However, there are some points that I would like the authors to address specifically in a minor revision of the manuscript:
- Fig. 3b: The box plots of L. camara and J. adhatoda look identical, please double-check the data here.

- In Fig. 3, please also add units for the y axes

- In the results section, lines 183ff: As you have provided sound files of the calls from the three different host plants, please also include a reference to these files in the results. Additionally, please include a description for the sound files indicating the recording location and the recording temperature.

- Paragraph starting at line 219: I think the lack of any discernible difference in the song parameters echeme duration and echeme period might not be due to morphological constrains of the song production aparatus but might be rather linked to those parameters being generally quite variable, depending maybe more on the animal's internal state than on mechanical constraints. See also my comments in this section in the attached pdf.

I have also marked a few minor points directly in the attached pdf.

---

## Round 0.3 · accepted · Accept

I am pleased to inform you that your revised manuscript titled "Delineating the calling pattern of Oecanthus indicus from native and non-native plant species" has been accepted for publication in PeerJ. The revisions you made in response to the reviewers' comments have significantly improved the quality and clarity of the manuscript.